# Biological Effects and Applications of Bulk and Surface Acoustic Waves on In Vitro Cultured Mammal Cells: New Insights

**DOI:** 10.3390/biomedicines10051166

**Published:** 2022-05-18

**Authors:** Agathe Figarol, Lucile Olive, Olivier Joubert, Luc Ferrari, Bertrand H. Rihn, Frédéric Sarry, Denis Beyssen

**Affiliations:** 1Institut FEMTO-ST, UMR CNRS 6174, Université de Bourgogne Franche-Comté, F-25030 Besançon, France; agathe.figarol@femto-st.fr; 2Institut Jean Lamour, UMR CNRS 7198, Université de Lorraine, CNRS, IJL, F-54000 Nancy, France; lucile.olive@outlook.fr (L.O.); olivier.joubert@univ-lorraine.fr (O.J.); luc.ferrari@univ-lorraine.fr (L.F.); bertrand.rihn@univ-lorraine.fr (B.H.R.); frederic.sarry@univ-lorraine.fr (F.S.)

**Keywords:** ultrasounds, surface acoustic waves, mammal cells, cytotoxicity, proliferation, migration, cell permeability, cell sorting, wound healing

## Abstract

Medical imaging has relied on ultrasound (US) as an exploratory method for decades. Nonetheless, in cell biology, the numerous US applications are mainly in the research and development phase. In this review, we report the main effects on human or mammal cells of US induced by bulk or surface acoustic waves (SAW). At low frequencies, bulk US can lead to cell death. Under specific intensities and exposure times, however, cell proliferation and migration can be enhanced through cytoskeleton fluidization (a reorganization of the actin filaments and microtubules). Cavitation phenomena, frequencies of resonance close to those of the biological compounds, and mechanical transfers of energy from the acoustic pressure could explain those biological outcomes. At higher frequencies, no cavitation is observed. However, USs of high frequency stimulate ionic channels and increase cell permeability and transfection potency. Surface acoustic waves are increasingly exploited in microfluidics, especially for precise cell manipulations and cell sorting. With applications in diagnosis, infection, cancer treatment, or wound healing, US has remarkable potential. More mechanotransduction studies would be beneficial to understand the distinct roles of temperature rise, acoustic streaming and mechanical and electrical stimuli in the field.

## 1. Introduction

Ultrasounds (USs) are widely used in the medical field, and increasingly in the wider area of biotechnologies. The most famous application is medical imaging, using frequencies from 1 to 10 MHz, and an intensity of lower than 720 mW cm^−2^ [1]. In this context, US shows, remarkably, no or negligible toxicity towards biological tissues. Low-intensity pulsed ultrasounds have been shown to enhance tissue repair and bone regeneration [2,3]. Other medical applications of US were researched. At low frequencies, a phenomenon called cavitation can create transient pores in the cell membranes and locally increase the delivery of therapeutic drugs through translocation [4]. At higher frequencies, transfection can be achieved without cavitation, facilitating gene or protein delivery with great application potential in oncology [5,6]. US can be triggered by the resonance of a whole bulk material (as shown in Figure 1A,B), or the resonance of the extreme surface of an elastic material (Figure 1C). These surface acoustic waves (SAW) are due to a piezo-electric system stimulated by an interdigital transducer (IDT). They allow for the microfluidic manipulation of very small volumes to single cells, and could enhance wound healing [7,8]. Studies of US physical and molecular mechanisms of action represent a rising field across physical and biological sciences.

In this article, we review all studies on the action of US with frequencies of over 1 MHz on human or mammal cells. Studies are split according to US frequency: first from 1 to 10 MHz; and over 10 MHz. The latter coincides with more recent works. Then, this review explores SAW, excluding standing SAW for concision, and to avoid redundancies, we include the latest articles on the subject [9,10,11]. Biological outcomes are questioned, as well as the physical phenomena that trigger them, including cavitation, mechanical stimulation, or acoustic streaming.

Before starting to review the literature in this field, we define some terms linked to US stimulation characterization. In Figure 2, the main parameters are outlined. The wave frequency (in Hz) is reciprocal to the period (in s). Stimulations are often in the pulse mode, with a duty cycle defined as the ratio of stimulation time (ON time) to total time (ON time + OFF time). The duty cycle is equivalent to the pulse period, thus reciprocal to the pulse-repetition frequency. The dose, expressed in J cm^−2^ or W s cm^−2^, is defined as the product of the intensity, expressed in W cm^−2,^ and the exposure time, expressed in s.

## 2. Review Method

### 2.1. Research Design

Objective: To review the scientific literature on the impacts of ultrasounds generated by acoustic waves on mammal cells, on in vitro models.

Inclusion criteria: peer reviewed articles or book chapters, referenced in scientific databases, in English, containing at least two of the keywords, no criteria of publication date.

Exclusion criteria: studies focusing on standing surface acoustic waves, of a poor scientific quality, or a studies that do not provide enough parameters for the comprehension and comparisons of its results.

### 2.2. Selection and Extraction of the Studies

Keywords: surface acoustic waves, acoustic waves, ultrasounds, cells*, bio*, (effects OR impacts). 

Data sources: We identified suitable studies by searching electronic databases and scanning reference lists of articles. We used Web of Science and Google scholar (last accessed date: 5 May 2022).

The selection of studies was performed in two stages: firstly, selection was carried out by two people (LO, DB), independently, to review any issues and with a focus on low frequency ultrasounds (<10 MHz). Then, two authors (AF, DB) independently assessed the eligibility of studies for secondary validation, with a greater focus on high frequency ultrasounds (10 to 1000 MHz) and ultrasounds induced by surface acoustics waves, as well as the inclusion of newer studies on the three topics. Any disagreements were settled by consensus. Other authors could suggest a particular study, if not yet selected, and the study was checked for compliance with the inclusion or exclusion criteria and selected accordingly.

### 2.3. Analysis of the Studies

Literature reviews were read and included as part of the discussion. The following three summary tables were built reporting the main parameters and results of the studies: one for the low frequency ultrasounds, one for the high frequency ultrasounds, and the third for ultrasounds induced by surface acoustic waves. For the last part, the electrical power to the IDT had to be extrapolated from the voltage (root mean square peak, or peak to peak) across the electrode in some cases, leading to the hypothesis that the electrode impedance is 50 Ω. There was no information about the electrical impedance of IDTs, but given the expertise in the field, the error made for the electrical power with this assumption should be small enough to consider the order of magnitude. Indeed, traditionally, some SAW devices have an electrical-impedance matching circuit in order to increase the energy transfer between the energy source and the SAW device. If there is no impedance matching, the standing wave ratio (SWR) of the IDT never exceeds 1.5, which means that 80% of the incident electrical energy is transmitted to the IDT and, therefore, 20% is reflected. An SWR of 1.2 to 1.3 is closer to reality for standard bi-directional electrodes, which translates into nearly 90% of the incident energy being available at the IDT. The error made by making this assumption (50 Ω) will, therefore, be only 10 to 20%, at most which is acceptable and allows for a good order of magnitude with which to compare the works.

## 3. Ultrasounds at Low Frequencies (<10 MHz)

The biological effects of US at low frequencies have been extensively studied as USs are extensively used for medical investigations [2,12,13,14,15]. The following sections provide a study of their potential impacts on human and mammal cells as a function of US frequencies and exposure time. Table 1 recapitulates those findings and was provided to assist in the reading of this review. 

### 3.1. Adverse Effects on Cells

US can trigger apoptosis and a low level of necrosis, as demonstrated for leukemic cells exposed to low frequencies of US generated by a ceramic disk (1.8 MHz frequency, 7 mW mL^−1^ intensity, exposure from 1 to 18 h) [19]. The hypothesis, proposed by Lagneaux or Miller et al., is that apoptosis is triggered by the presence of the ^1^O_2_ oxygen singlet, in an instable and highly reactive state of dioxygen, due to the sonoluminescence caused by a cavitation phenomenon in the medium [19,30].

Genotoxicity appears to be another effect of this inertial cavitation phenomenon, either linked to oxidative stress or to the mechanical constraints of the cavitation alone. DNA double-strand breaks caused by these forces were evidenced in leukemic cells exposed to US at a 1 MHz frequency, with a 10% duty cycle, a 100 Hz pulsed wave, an intensity higher than 200 mW cm^−2^, and an acoustic pressure of higher than 0.105 MPa [17]. Nevertheless, Udroiu et al. demonstrated that US can affect the genome integrity even at intensities below the cavitation threshold [29]. Transient mitotic anomalies were observed after a 30 min US stimulation at 1 MHz, and an intensity of either 70, 140, or 300 mW cm^−2^ defined by the authors as, respectively, below, around, or over the cavitation threshold. This genotoxic effect was also observed in the following cell types: HeLa human cervical cancer cells, MCR-5 human pulmonary fibroblasts, and MCF-7 human breast cancer cells.

Adverse outcomes from US exposure may, however, depend on the cell type. Lagneaux et al. revealed that cancerous cells seemed to be more sensitive to US-induced necrosis than non-cancerous cells [19]. Other researchers have studied how to induce selective cell death. Narihira et al. studied the effects of US in the presence or absence of Cetuximab (an anticancer drug)-coated albumin microbubbles on oral squamous carcinoma cells (HSC-2 cells) and tumor monocytes (U-937) [23]. The cells were exposed to a US of 1 MHz, with a 10 Hz repetition-pulse frequency, and a duty cycle of 50%. Intensities of 0.8, 0.9, and 1 W cm^−2^ were delivered for 15 s, which corresponded to 150, 160, and 170 kPa pressures. Whatever the intensity, the viability decreased in a dose-dependent manner in HSC-2 cells only.

### 3.2. Proliferation, Cytoskeleton Rearrangement and Transfection

When the parameters are properly calibrated, US can enhance cell proliferation and migration. Studies have investigated the occurrence of wound healing and bone regeneration. Using acoustic intensities from 30 to 1000 W cm^−2^ and frequencies between 1 to 3 MHz, US could positively affect the differentiation and protein synthesis of osteoblasts, osteoclasts, chondrocytes and mesenchymal stem cells [2]. At 1 MHz, 250 W cm^−2^, and a duty cycle of 20%, the proliferation rate of murine osteoblasts increased by 20% [21]. The speed of the scratch-wound healing increased with US stimulation even when the proliferation was blocked with mitomycin C, hence, authors observed increased migration as well as proliferation with US. The parameters for a maximal proliferation seem, once again, to depend on the cell types. On murine myoblasts (C2C12), the most efficient parameters to increase proliferation were 3 MHz and 1 W cm^−2^ (20% duty cycle, negligible medium heating), but were 1 MHz and 500 mW cm^−2^ for differentiation [25]. On rat pheochromocytoma cells (PC-12), however, if a 138 to 186% increase in proliferation was noted, no significant difference between the stimulation parameters was observed. The frequency used was 1.48 MHz, the maximal pressure was 45 kPaw with a 15, 30, 50, 70% duty cycle and 5, 10, 20, 30 min stimulation three times a day for three days [20]. In another study, at 1 MHz and 0.1 to 1 W cm^−2^, a significant rise in primary osteoblasts and fibroblasts proliferation was also observed (47% or 37% at 0.7 or 1 W cm^−2^ for osteoblasts, and 34% or 52% for fibroblasts) [16]. Interestingly, the collagen synthesis rose as well at 0.1 to 0.7 W cm^−2^ or 0.1 to 0.4 W cm^−2^ for fibroblasts and osteoblasts, respectively. 

The impacts of US on the cytoskeleton and proliferation were questioned in several recent studies. Raz et al. hypothesized that cell sonication induces transient alterations leading to cytoskeleton reorganization, cell proliferation and migration (Figure 3, top) [24]. Those effects were linked to mechanical energy transfer to the cells, increasing as a function of US frequency until reaching a plateau. A 60% increase in cell proliferation was evidenced in bovine endothelial cells following 15 to 30 min 1.2 W cm^−2^ US stimulation with a frequency of either 0.5, 1, 3.5, or 5 MHz, and a duty cycle of 50 and 100%. At 15 min, a difference in cell proliferation was observed between the duty cycle of 50 and 100%, but this disappeared at 30 min. Moreover, the study underlined morphological changes in actin fibers and the disassembly of their focal-adhesions and microtubules (Figure 3, top). Initial states were recovered after 24 h, supporting the authors’ hypothesis. Focal adhesions are constituted mainly by integrin, which has been shown to be activated by low-intensity pulsed ultrasounds [3]. These effects on the organization of the cytoskeleton and cell proliferation appear to be a function of the cell types. Indeed, Schuster et al. demonstrated that for an equivalent US dose, no impact on actin and focal adhesions was observed, but an increase in proliferation was apparent for the human cardiac microvascular endothelial cell line (hcMEC) [27]. Moreover, the proliferation rate of the Madin–Darby Canine kidney epithelial cell line (MDCK) increased with the US energy until 25 W s cm^−2^, after which it began to decrease. For a mouse neuroblastoma line (Neuro2A cells) or the human colon adenocarcinoma cell line (HT29), the proliferation rate increased only at a high energy (600 W s cm^−2^) and dropped at lower energies. In addition to the proliferation rate, electronic microscopy showed an increased number of cells presenting plasma membrane blebs, which might be a sign of apoptosis (Figure 3 bottom). Using a similar protocol, a second study showed an increase in neural stem cell proliferation, but no impact on neurogenesis and gliogenesis [28]. 

Other studies sought to better understand this phenomenon of cytoskeleton disorganization, and showed its “fluidization” under US stimulation. Fluidization is a phenomenon where soft materials change from a solid to a fluid-like state when subjected to shear stress [31]. In cell biology, the so-called cytoskeleton fluidization indicates a reorganization of the actin fibers and microtubules, leading to deformations of the plasma membrane. In the study of Mizrahi et al., the cytoskeleton of human Airway Smooth Muscle cells (HASM) showed fluidization under US stimulation at 1 MHz. Following exposure to US at 1 W cm^−2^ and 20% duty cycle (to minimize the temperature rise to under 1 °C), the effects were transient, and a repolymerization of the actin filaments was observed within 200 s. At 2 W cm^−2^, however, the effects were irreversible and US led to cell death. The fluidization could be due to the compression waves that generate the local deformation of the cell [22]. Samandari et al. developed a simulation model and compared it to their experimental outcomes. Their standard linear solid viscoelastic model showed that cell deformation increases with increased pressure. These deformations might depolymerize the actin filaments and activate signaling pathways sensitive to mechanotransduction. The deformations are more important when the cell is spread out and close to the substrate. In their experiment, C2C12 cells were stimulated with US at a 0.8 or 1.7 MHz frequency, generating a pressure of 150 or 250 kPa, for 10 to 30 s. The temperature rise remained below 1 °C. No cavitation was observed. Cell death increased with pressure and frequency, even though it remained below 15%. Microscopic observations showed, as expected, a rearrangement of the actin cytoskeleton and blebs formation [26]. No studies have yet underlined the effects of US on the cytoskeleton intermediate filaments, including vimentin, keratin, lamin, desmin, etc. 

As a consequence of the effects on the cell cytoskeleton, US can temporarily disrupt the cell membrane. This property was used for transfection, i.e., the controlled introduction of exogenous genetic material such as genes or proteins into a cell (Figure 4) [4]. As a proof of concept, plasmid DNA was transfected to HeLa cells exposed to US at 1 MHz, 300 mW cm^−2^, a 50% duty cycle, and 5 Hz pulsation frequency [18]. 

### 3.3. Towards an Understanding of the Physical Mechanisms of Action

Several teams tried to formulate hypotheses and propose models to explain the physical phenomena at play in the biological effects of US. The resonance and shear stress forces could provoke disjunctions between molecular complexes or conformational changes of biomacromolecules. Indeed, the hypothesis presented by Johns suggests that the absorption of US energy by enzymes could lead to their activation. The link between an enzyme and its inhibitor may be broken, or the enzyme may adopt an active conformation on its own. In both cases, the biochemical reactions that the enzyme catalyzes are boosted [32]. Other biomacromolecules could be affected, such as the lipids forming the cell membrane. A study, published in 2011, suggested that the US mechanical energy impacts the hydrogen bonds between the two phospholipid layers of the plasma membrane and transforms them by contracting and expanding the intramembrane space [33]. These constraints could explain the cytoskeleton reorganization, and eventual potential membrane disruption, with irreversible impacts at high frequencies. The cavitation and ensuing microbubbles formed in the culture medium might act as amplifiers of the phospholipids’ reorganization. 

Cumulative effects of the stress impacted by the resonance on the organelles could also lead to a fatigue phenomenon, which explains the observed cellular damage. Kimmel [34] developed a model to understand the impacts of US on cell membranes, without thermal and cavitation effects. Frequencies varying from 0.001 to 100 MHz were applied to objects of 100 nm, 1 µm and a 5 µm radius. The following four rheological models were tested: viscous fluid, elastic solid, and Voigt and Maxwell viscoelastic constructs. It was shown that the resonance frequency, the frequency for an intracellular vibration of maximal amplitude, was radius dependent. Furthermore, 100 nm radius objects, similar in size to cell organelles, resonated at 1 MHz, a frequency that is currently used for medical applications. Miller et al. confirmed such findings for chondrocytes (12 µm radius) with maximal deformation regardless of the pressure of US with a resonance frequency of 5.2 MHz. At other frequencies, the deformation increased with the pressure but to a lesser extent [30]. 

The impacts of US depend on physical parameters such as resonance frequency and acoustic pressure, but also on biological parameters such as the cell size, adherence, and type. The frequency, pressure or dose units are not sufficient to comprehend the US effects on cells. A review from 2007 [35] stated that mW cm^−2^ the most frequently used intensity unit, even if simple and easy to apprehend, does not explain the acoustic field characterization at the studied area. The acoustic shear was rarely taken into consideration in the reviewed studies. Nevertheless, these studies present avenues by which to better understand the physical phenomena at play for low frequencies, or as seen in the next paragraph, for frequencies higher than 10 MHz.

## 4. Ultrasounds at High Frequencies (10–1000 MHz)

At high frequencies, the cavitation phenomenon is not observed; moreover, the beamwidth becomes narrower, allowing for a more precise cell stimulation. Technologies using US at frequencies of higher than 10 MHz were developed recently, such as single-cell imaging [36,37]. We will focus here on the direct impact of applications of US in the cellular or medical sciences on cell behavior, as summarized in Table 2.

### 4.1. Activation of Ion Channels, Applications in Oncology and Neurostimulation

Studies have shown that the permeability enhancement by US at high frequencies, or high frequency microbeam stimulation (HFUMS), seemed to depend on the invasive nature of the cells. Hwang et al. showed that US at 200 MHz increased cell permeability more significantly for human breast non-cancerous cells compared to cancer cells, as evidenced by Rhodamine B reflux [39]. A higher voltage induced a higher impact on permeability. Further studies concluded that HFUMS can enhance cell permeability through the activation of specific ion channels [40,41]. Ion channels are membrane proteins that facilitate the transport of a specific ion or a family of ions down the electrochemical gradient (see Figure 5). They are ubiquitous, crucial for the physiology of excitable cells, especially neurons, and their activity is modified in cancerous cells. A significant difference in Ca^2+^ influx was indeed observed following exposure to US at 193 MHz of human breast cancer or non-cancerous cells, and US of 200 MHz to endothelial cells (HUVEC) [40,41]. Likewise, another study found no impact of US at 50 MHz on human breast non-cancerous cells, but an increase in Ca^2+^ influx, as a function of the invasiveness of human breast cancer cells [5]. Another class of cationic channels, the transient receptor potential (TRP) channel, could also play a part, but no significant activation by US was observed. All the studies relied on the fluorescence index as a sensor of Ca^2+^ concentration changes. A transcriptomic analysis of the genes involved in the piezo channel or the TRP channel is welcomed. The detection of the differences in cell response towards HFUMS could distinguish between non-cancerous and highly invasive cancer cells. Moreover, this kind of stimulation by US showed no impact on cell viability, displaying optimal parameters for potential applications of HFUMS as biosensors [5]. In addition to being a tool for diagnosis, HFUMS could assist in tumor treatment. Daily exposure, for a 30 min period, to HFUMS in combination with light-emitting diodes (LED) induced a significant decrease in the proliferation of human cervix carcinoma cells (HeLa) [38,44]. This effect was shown for a frequency of 15 and 100 MHz, and at a range of intensity of higher than 100 W cm^−2^. Similar proliferation drops were found at 100 MHz with US only (no LED) [45]. The authors supposed this could provide a new avenue for cancer treatment. 

Prieto et al. [42] conducted further work on the activation of ionic channels by HFUMS in hope of developing applications in neurostimulation, and treatments against mental and neurological disorders (Prieto et al., 2018). The study used Chinese hamster ovary cells (CHO) modified to express mouse piezo 1 channel, or rat Na_v_1.2 channel, which is a type of sodium channel, or human embryonic kidney cells (HEK) modified to express mouse piezo 1 channels. The cells were exposed for 0.7 s to US at a frequency of 43 MHz, and an intensity of 50 or 90 W cm^−2^. This work confirmed the activation of the piezo channel by the US, more specifically by the acoustic radiation pressure and streaming. The Na_v_1.2 channel was also activated by the stimulation, albeit only due to thermal heating. Heating can indeed activate or increase the kinetic speed of ion channels. It has to be noted that the temperature rise at play here was only of 0.8 °C. Thus, even a small difference in environmental temperature due to US could impact the cell response.

### 4.2. Increase in Permeability and Transfection

Given the US effects on the membrane permeability through channel activation, the HFUMS can additionally be used to transfect small molecules, DNA plasmids and RNA messengers. At high frequencies, no microbubble is needed. HFUMS thus facilitates the controlled and local intracellular delivery of chosen molecules. A US-transfection system was developed by Yoon et al. at 150 and 215 MHz and tested on HeLa cells. The size and amount of transfected fluorescent dextran molecules depended on the frequency, the number of electric pulses, the peak-to-peak voltage (V_pp_), and pulse duration (t_p_). The study optimized the parameters for a maximal transfection of 3 kDa dextran molecules, without any significant impact on cell viability in the short (6 h) and long (40 h) term. The optimized parameters were as follows: V_pp_ = 22 V and t_p_ = 30 µs, or V_pp_ = 43 V and t_p_ = 10 µs [6]. A year later, the same team used this method to successfully transfer CRISPR-Cas9 systems and succeeded in reprogramming the genome of HeLa cells [43]. HFUMS-transfection was thus confirmed as an efficient technique for efficient genome editing. 

## 5. Ultrasounds Induced by Surface Acoustic Waves

The previous sections focused on bulk acoustic waves, where the whole transducer resonates to produce ultrasounds in the environmental medium. Here, we study the impacts of surface acoustic waves (SAW), where only the extreme surface of an elastic material resonates. The SAW are also called Rayleigh waves, in reference to the name of the first scientist to describe them in 1885. The generation of SAW requires the conversion of electrical energy to mechanical energy. A voltage is applied to a metallic interdigitated transducer (IDT) on the surface of a piezoelectric substrate, generally a lithium niobate (LiNbO_3_) chip, on which acoustic waves propagate longitudinally (Figure 1 bottom). These acoustic waves can propagate to other adjacent media such as the cell culture plate and culture medium, in which they create an acoustic streaming. Cells can thus be directly stimulated by mechanic waves or by shear flow. Due to the small size of these microsystems and their relatively low cost, SAW have a wide range of cell manipulation applications, refining and completing those of bulk acoustic waves (Table 3) [46,47].

### 5.1. Controlling Cell Detachment

A time-dependent detachment of human cells from their growing substrate can be observed when exposed to SAW. Likewise, Stamp et al. applied a power of 300 to 500 mW to a LiNbO_3_ chip, inducing SAW and US (no information was provided about their frequency and intensity) that detached adherent human osteosarcoma sarcoma osteogenic cells (SaOs-2) [58]. They hypothesized, however, that this loss in cell adhesion was due to an increase in temperature or a decrease in the pH of the medium, not the SAW and US. Indeed, when the environmental temperature was maintained under 37 °C, no significant cell detachment was observed. In order to control the temperature, the duty cycle can be decreased, and the number of exposure cycles can be increased to deliver an equivalent dose to the cells in a pulse rather than a one-time exposure. However, a recent study showed that changing the number of exposure cycles had no effect on the observed cell detachment for a similar exposure time and applied voltage [59]. Part of the remaining adherent cells in this study was destroyed through excess shear. Jötten et al. previously demonstrated that the shear flow also impacted the cell-detachment rate [52]. Other parameters are also involved such as the cell density, the cell type, rigidity and invasiveness (etc.).

A study described different behaviors before the detachment of red blood cells (RBC) depending on their pathological state [56]. For non-treated, non-infected RBC, the cell membrane was translated and either rolling or flipping occurred across the substrate before detachment. Glutaraldehyde-treated RBC showed a similar behavior but required a longer period of time before detachment. Malaria-infected cells, on the other hand, adhered quite strongly for the duration of SAW exposure, even if some cell translating, rolling and flipping was observed. These findings pave the way for potential applications of SAW in diagnosis with the use of biomarkers linked to mechanotransduction. 

### 5.2. From Cell Manipulation to Cell Sorting 

Single-cell manipulation is a rising field at the intersection of biological sciences, microfluidics, and acoustics. SAW can be used to facilitate cell collision with nanoparticles and to induce cell lysis in very small volumes in microfluidic systems [60]. SAW can also guide cell seeding further into porous scaffolds than non-exposed cells [53,61]. A two-dimensional cell-seeding pattern may also be built with possible spatial single-cell isolation [49]. Once isolated, single cells can be analyzed. Collins et al. showed that the US velocity provided information on the elasticity of the cells with 10^6^ more sensitivity than atomic force microscopy (AFM). SAW could thus be used for the differentiation of cancer and healthy cells, as the elastic modulus might be a possible biomarker for invasiveness or metastatic potential. 

A high rate of cell manipulation and sorting can be achieved. A study in 2014 reached a sorting rate of as high as 3000 cells s^−1^ [54]. Under the influence of SAW, cells were sorted depending on their fluorescence. The fluorophore (calcein-AM) used was sensitive to cell metabolic activity and membrane integrity. In the same year, a study realized the sorting of red blood cells by acoustic streaming depending on their infection states by the malarial parasite *Plasmodium falciparum* [48]. The authors noted that the cells’ density impacted their displacement within the shear flow. Of note, powers of above 250 mW did not lead to significant differences in cell behavior, while powers of as low as 65 mW promoted efficient cell sorting. More recently, an attempt was made to first sort fluorescent polymer beads depending on their size, then brain cancer cells depending on their size and virulence [8]. Separation increased with the SAW cycle number. SAW induced more stability and flexibility in the cell sorting than standing SAW. Importantly, at the power (126 mW) and frequency (39 MHz) used, the authors detected no significant effect on cell viability, proliferation and migration. 

### 5.3. Wound Healing: Cell Migration or Proliferation?

In 2016, SAW were seen to enhance wound healing, with cells exposed to low powers (2 to 4 mW, at 159 MHz) for 48 h [57]. Osteoblasts (SaOs-2) were seeded as a monolayer with a zone of a few cm left empty, representing the “wound”. After stimulation, the cells recreated the monolayer at a faster rate to join both sides of the wound, leading to the so-called “healing” process. Increasing the US intensity seemed to increase the healing process. No significant necrosis of the cells was observed. It remained unclear if the effect was due to an increase in cell migration or proliferation, and to mechanical or electrical stimulation. No preferred direction of migration or proliferation was detected; thus, the shear stress was suggested to be responsible for the wound-healing intensification. 

In 2020, Brugger et al. conducted a similar experiment, and confirmed the improvement in wound healing using SaOs-2 and canine and human kidney cells [7]. Here, again, there were no morphological changes or oxidative stresses detected. This was only observed, however, if the flow stream was at a reasonable level; if the flow stream was too strong, cell detachment was observed, which coincides with earlier findings described in Section 4.1 in this review. The experiment found that both cell migration and proliferation were enhanced, with the predominance of cell migration. Direct mechanical stimulation seemed to have more of an effect than electrical stimulation, but further studies are needed to confirm this hypothesis. The rise in temperature needs to be controlled with an observed increased dependent on the used power: ΔT/ΔP = 37 K/W. Recently, Imashiro et al. reinforce Brugger’s findings on cell migration, with a SAW system where the temperature was controlled and maintained between 36 and 38 °C, and the electrical stimulation was negligible as isolating layers of glycerol and PDMS separated the cells from the chip [59]. An increase of 28 and 42% in the cell migration speed was observed at 2 and 4 V, but the migration was suppressed at 18 V, which corresponds to a 59.3 mW cm^−2^ intensity. In contrast to Stamp’s study, they found a significant preferential alignment in the cell nuclei. They suggested it to be linked to changes in the cytoskeleton, specifically to an increase in actin stress fibers and bundle thickness. The shear stress, estimated to be 3.7 mPa, was thought to be too low to induce such biological impacts and rather, they were attributed to the propagating acoustic waves themselves [51]. 

The question of whether cell proliferation could be enhanced by SAW is especially pertinent when compared to the impacts of bulk US (Section 2.2). SAW could increase human monocyte proliferation by up to 36% using the following parameters: 49 MHz, 467 mW, duty cycle of 2.5% and 48 h exposure [50]. The temperature rise played no role, as they reduced the heating to under 0.5 °C using a pulse stimulation with a 2.5% duty cycle. In this study, however, the authors postulated that the acoustic streaming rather than the mechanical stimulation was responsible for cell proliferation. Lower shear stress, without US, has indeed been seen to increase the production of F-actin in human monocytes, inducing structural changes of the cytoskeleton that could lead to an increase in proliferation. Considering their wide use and important potential applications, new studies on cell mechanotransduction activated by SAW are expected in the next few years.

## 6. A Need for Experimental Standardization

The study of the acoustic wave’s effects on human and mammal cells is still developing. One major drawback is the lack of standardization between the published works. A primary inconvenience is indeed the difference in US parameters, of either the intensity, the power, or the voltage, especially in the most recent articles on HFUMS or SAW. Considering the wide range of used cell types, a comparison of results from different studies is quite challenging. As shown in Figure 6, an attempt was made to summarize the tendencies described throughout this review. Low-frequency acoustic waves were seen to induce cell death due to cavitations in the culture medium, which, if used with optimized parameters, can perturb the cell membrane just enough to ease gene or protein translocation with the help of microbubbles. Low-frequency US also mainly affects the cytoskeleton, with its fluidization and the formation of blebs. It can either be irreversible and induce cell death, or reversible and enhance cell proliferation and regeneration with applications in tissue regeneration. This is likely due to acoustic pressure and resonance frequencies close to those of the cell components. At a high frequency, the cavitation phenomenon is not observed, but the membrane permeability can still be enhanced with applications in oncology, neurostimulation, or the transfection of genetic material. Several studies demonstrated the stimulation of piezo ionic channels. Once the mechanical transfers of energy are understood, much work is needed to assess the roles of temperature and acoustic shear. SAW are mostly used for their potency in cell detachment, cell sorting and wound healing by increasing cell proliferation and/or migration. The cellular mechanisms of SAW-induced US depends on the cell density and on properties such as the size or the elastic modulus. For example, this could facilitate the detection and separation of infected red blood cells. Most of these microfluidic manipulations are linked to the acoustic shear flow, as they occurred under a stabilized temperature of the cell environment. However, we have seen that even a slight change in temperature could impact the ion channel activation, thus, further study on this parameter is required. Moreover, the role of mechanic and electric stimuli in the biological response of the exposed cells have yet to be clarified. 

Numerous parameters influence these results, however, including cell type, concentration, pathological state, US frequency, intensity, pulse mode, exposure time, and more globally, the dose of exposure, as well as the environmental temperature and shear flow. Large-scale studies focusing on only one parameter would help to develop an understanding of the biological and physical mechanisms of action. This kind of work would contribute to experimental standardization across laboratories, an efficient workload sharing, and the dissemination of systematic results. 

## 7. Perspectives in Biomedicines

In the introduction, conventional medical uses of acoustic waves were discussed, with applications in imaging and bone repair. Other forms of imaging such as electrography and ultrasonography rely on ultrasound-induced impacts too, such as the stiffness or elasticity of the targeted tissues. Lithotripsy employs high-amplitude acoustic shock waves to break kidney stones. It has cosmetic applications too, more specifically for laser-tattoo removal [51]. The effects rely on ultrasound-induced tensile strengths exceeding the fracture threshold for graphite, and acoustic waves impacting the environmental cavitation bubbles [62]. Treatments for musculoskeletal disorders have also been studied. For example, two clinical studies were launched that used low-intensity and low-frequency surface acoustic wave ultrasounds for nervous tic (from trigeminal neuralgia)-relief treatment [63,64].

In the following paragraph, biomedical applications are discussed with regard to recent advances in understanding the cellular effects of the bulk or surface acoustic waves. For example, high-frequency ultrasounds could enhance medically-assisted reproduction success rates. Surface waves derived from 19 MHz and 2 W stimulation could increase sperm velocity by up to 34% [65]. The underlying biological reasons could either be an enhancement in the metabolic rate, regulations of mechanosensitive ion channels, or changes in the membrane stiffness which influence the rate of energy dissipation into the fluid. Potential membrane permeability modifications, or improved flow velocity, could also increase tissue oxygenation. In a clinical study, patients with ischemic feet were treated with 0.1 MHz of SAW for 30 min. An increase in oxygen saturation was indeed observed [66]. However, perhaps the most promising biomedical field for acoustic waves is oncology. Choi et al. conducted a feasibility study on the suppression of tumor-cell proliferation, as previously discussed [44]. One of the main challenges is the selectivity of the process. Cell manipulation with SAW could assist in the separation of cancer cells from the healthy population, but as of yet, this has only been demonstrated in vitro for circulating cells [67]. Focused ultrasound surgery is a novel noninvasive technique that induces necrosis by increasing the temperature at a precise point in the body. The acoustic power varies according to the tissue nature and depth, with values from 100 to 200 W at the skin line and from 500 to 700 W cm^−2^ at the focal point with 1 to 1.5 MHz and for <2 h. Clinical studies for uterine fibroid tumors, or prostate cancer, showed its pertinence [68,69,70]. High-intensity focused ultrasounds could also be applied for noninvasive brain therapies, with localized changes in transgene expression [71]. Other promising new therapeutic applications of ultrasounds are only at the preclinical stage, or represent in vitro investigations. In this review, studies using high-frequency acoustic waves to activate ion channels leading to an increase in cell permeability, were discussed with regard to applications in oncology, neurostimulation, or in the delivery of drug and genetic materials. Clinical applications of SAW for wound healing are expected in the next few years. In parallel, the development of non-cell-based biomedical uses of acoustic waves such as biosensing or antibacterial treatment could lead to innovative combined applications.

## 8. Conclusions

This review summarized the biological outcomes of exposing human or mammal cells to US, and provides leads on the physical phenomena at stake. Even if it is not as emphasized with SAW, US can lead to cell death, damage, or a decrease in their proliferation if the parameters are not well adjusted. Another point of interest for cellular or medical sciences is the possibility of triggering an increase in cell proliferation, migration, and permeability either by inducing changes in the cytoskeleton or ion channel activity. Overall, standardized studies to assess the impact of each physical parameter will be conducted to anticipate specific cell-line responses to US and to design efficient microsystems for biomedical applications of acoustic waves.

## Figures and Tables

**Figure 1 biomedicines-10-01166-f001:**
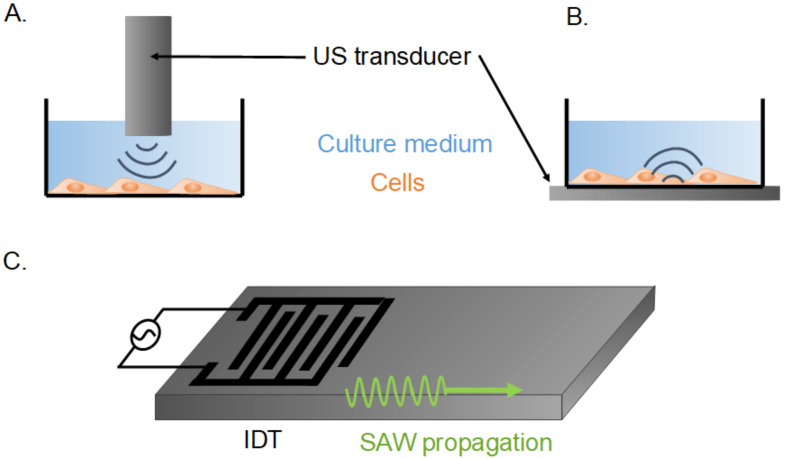
Schematic view of cell-stimulation systems of low- or high-intensity ultrasound stimulation. (**A**): Cells stimulated mainly by the shear flow induced by a US transducer immersed in the culture well. (**B**): Cells stimulated mainly by the mechanical vibrations of the culture well. US stimulated by the US transducer under it. (**C**): Piezo-electric system with an interdigital transducer (IDT) inducing surface acoustic waves (SAW).

**Figure 2 biomedicines-10-01166-f002:**
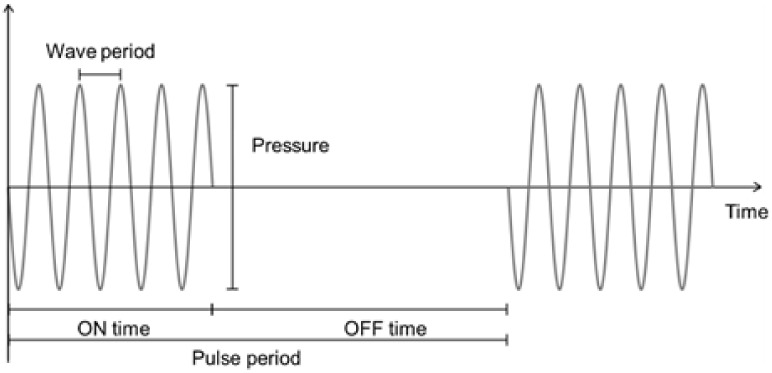
Graphical representation of the parameters defining ultrasounds.

**Figure 3 biomedicines-10-01166-f003:**
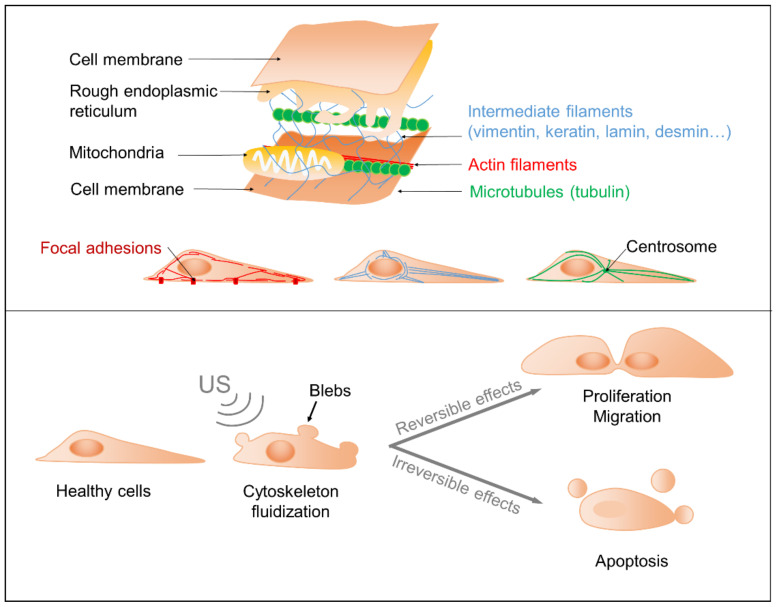
Cytoskeleton and main impacts from US exposure. Top: Schematic outcomes of US on cellular cytoskeleton, proliferation and migration. Bottom: Schematic visualization of the cytoskeleton components. Focal adhesions are integrin-containing multi-protein structures binding actin filaments to the extracellular substrate.

**Figure 4 biomedicines-10-01166-f004:**
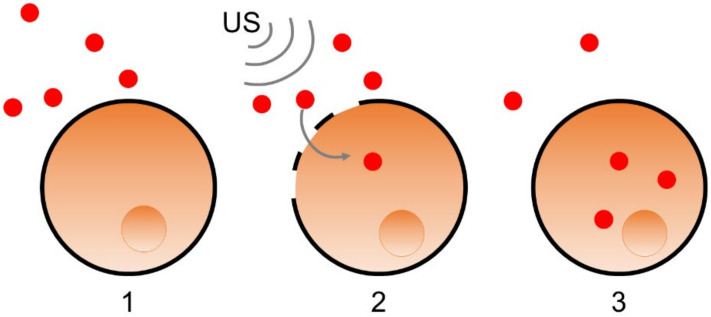
Schematic description of gene or protein transfection (schematized in red dot) into a cell (schematized in orange, with its nucleus in darker orange). The elements to be transferred are in the extracellular medium (1). The cell membrane is disrupted by US (schematized by the grey waves) (2). The cell membrane closes again after integration of the transfected elements (3).

**Figure 5 biomedicines-10-01166-f005:**
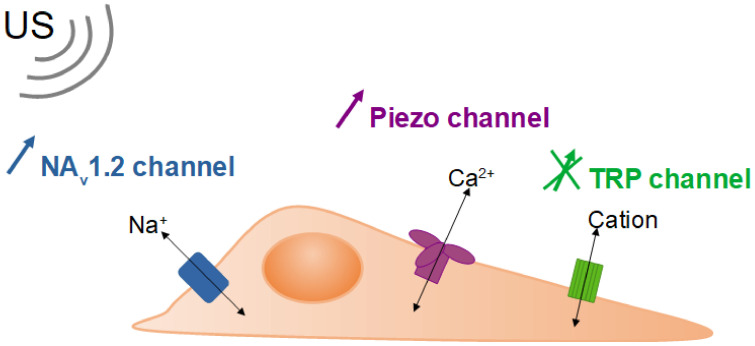
Schematic description of cellular ionic channels: Na_v_1.2 and piezo channels were shown to be activated by SAW, while no significant impact on TRP (transient receptor potential) channel was observed.

**Figure 6 biomedicines-10-01166-f006:**
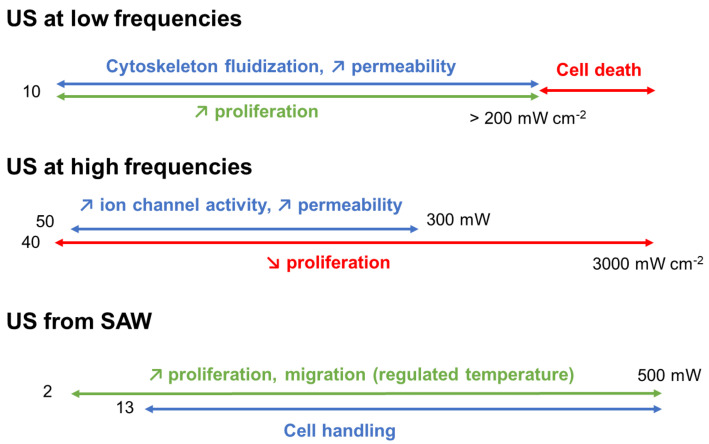
Sum up of the global tendencies of US effects on human and mammalian cells. Red: deleterious effects, blue: effects on the cytoskeleton and cell membrane, green: favorable effects.

**Table 1 biomedicines-10-01166-t001:** Summary table of the impacts on mammal cells of US at frequencies under 10 MHz. (↗: increase in, ↘: decrease in, N.A.: not available).

Reference	Frequency (MHz)	Intensity or Pressure	Duty Cycle (%)	Pulse Time (min)	Dose (J cm^−2^)	Cells	Temperature Control	Biological Effects	Hypothesis
[16]	0.045, 1	10–400 mW cm^−2^	25	5	7.5–75	Primary fibroblasts Primary osteoblasts Primary monocytes	Rise ≤ 1.8 °C	↗ proliferation ↗ collagen synthesis	N.A.
[17]	1	100–400 mW cm^−2^	10	1	0.6–2.4	Human monocytes (U-937) T lymphoblasts (Molt-4) Lymphocytes (Jurkat) Leukemia cell line (HL 60)	Rise ≤ 1 °C	↗ DNA double strand breaks if I > 200 mW cm^−2^	Free radicals formation, due to cavitation.
[18]	1	300 mW cm^−2^	50	0.5–15	4.5–135	Human adenocarcinoma epithelial cells (HeLa)	None	↗ membrane permeabilization ↗ intracellular transport	N.A.
[19]	1.8	7 mW mL^−1^	65	0.33	91 J mL^−1^	Human leukemia bone marrow cells (K562, KG1a) HL-60, human B cell precursor leukemia cells (Nalm-6)	None	↗ apoptosis Mild necrosis Virulent leukemic cells more sensitive	Oxygen singlet formation, due to cavitation.
[20]	1.48	0.045 MPa	15–70	5–30	N.A.	Rat pheochromocytoma adrenal medulla cells (PC-12)	None	↗ proliferation	N.A.
[21]	1	250 mW cm^−2^	20	30	90	Mouse osteoblasts (MC3T3-E1)	Pre-heated water tank	↗ proliferation ↗ migration	N.A.
[22]	1	1000–2000 mW cm^−2^	20	0.5	6–12	Human aortic smooth muscle cells (HASM)	Rise ≤ 1 °C	Reversible fluidization for I = 1000 mW cm^−2^ Damages to the actin filaments for I = 2 W cm^−2^	Fluidization due to the compression wave causing a local cell deformation
[23]	1	800–1000 mW cm^−2^	50	0.25	6–7.5	Human oral squamous carcinoma cells (HSC-2) U-937	None	↘ HSC-2 viability with microbubbles. No effect on U-937. No effect without microbubble.	N.A.
[24]	0.5, 1, 3.5, 5	1600–2000 mW cm^−2^	10–100	30	288–3600	Endothelial cells	Measured temperature “excluded the possibility that thermal effects may cause changes in the cultured cells”	↗ proliferation ↗ cytoskeleton disorganization ↗ tissue repair.	direct mechanical action
[25]	0.5, 1, 3, 5	250–1000 mW cm^−2^	20	5	15–60	Mouse myoblasts (C2C12)	Room temperature (28 °C) water tank	↗ proliferation ↗ differentiation	Mechanical constraints
[26]	0.8, 1.5	150, 250 kPa	100	0.17–0.5	N.A.	C2C12	Rise ≤ 1 °C	Induce cytoskeleton fluidization ↗ cell mortality	Cell deformation with acoustic pressure
[27]	0.51, 0.994, 4.36	N.A.	N.A.	N.A.	3, 25, 50	Human cardiac microvascular endothelial cells (hcMEC) Madin–Darby Canine Kidney cells (MDCK) Mouse neuroblastoma cells (Neuro2A) Human colon cancer cells (HT29)	Perfused water tank at 37 °C	↗ proliferation at low I Not anymore at high intensity	N.A.
[28]	0.51, 4.36	N.A.	N.A.	N.A.	3, 25	Neural stem cells	Perfused water tank at 37 °C	↗ proliferation no increase in neurogenesis or gliogenesis	N.A.
[29]	1	70–300 mW cm^−2^	100	30	126–540	HeLa Human fetal lung fibroblasts (MCR-5) Human breast cancer cells (MCF-7)	Rise ≤ 1 °C	↗ mitotic abnormalities as a function of I disassembly of focal adhesions and microtubules.	N.A.

**Table 2 biomedicines-10-01166-t002:** Summary table of the impacts on mammal cells of bulk US at frequencies from 10 to 1000 MHz. (↗: increase in, ↘: decrease in, N.A.: not available, *: extrapolation based on the hypothesis that the electrode impedance is at 50 Ω).

Reference	Frequency (MHz)	Voltage, Intensity or Electrical Power	Duty Cycle (%)	Pulse Time (s)	Dose (J cm^−2^)	Cells (Adherent)	Temperature Control	Biological Effects	Hypothesis
[38]	15 + LED	47.9, 82.15, 128.11 mW cm^−2^	100	1800 (daily)	126,000– 230,600	Human cervix carcinoma cells (HeLa)	None	↘ proliferation	N.A.
[39]	200	16, 32, 47 V 110, 230, 330 mW *	2.5	10	N.A.	Human breast cells (MCF-12F) Human breast cancer cells (MDA-MB-435)	Thermally controlled chamber	↗ cell permeability higher in non-cancerous cells	N.A.
[40]	200–1000	4, 8, 16, 32 V 30, 60, 110, 230 mW ***	0.0025–1	0.3–150	N.A.	Highly invasive human breast cancer cells (MDA-MB-231) Weakly invasive human breast cancer cells (MCF-7, SKBR3, and BT-474)	None	↗ Ca^2+^ influx as a function of invasiveness	N.A.
[41]	193	1.8–3.6 MPa	0.1, 0.25, 0.5, 0.75, 1	0.5	N.A.	Endothelial cells (HUVEC)	Thermally controlled chamber	↗ Ca^2+^ influx	N.A.
[42]	43	50,000, 90,000 mW cm^−2^ 3.2, 5.7 mW focused on 1 cell	100	0.7	35, 63	Chinese hamster ovary cells (CHO) expressing rat Na_v_1.2 or mouse piezo 1 channels Human embryonic kidney cells (HEK) expressing mouse piezo 1 channels	Estimated rise of 0.8 °C	Stimulation of the Na_v_1.2 and piezo channels	US through acoustic radiation and shear stimulate the piezo channel Thermal heating stimulates the Na_v_1.2 channel
[5]	50	0.43–1.97 MPa	33	3.3	N.A.	Human breast cells (MCF-10A) MDA-MB-231 MCF-7	Rise ≤ 0.5 °C	↗ Ca^2+^ influx, as a function of invasiveness	US stimulate the piezo channel
[6]	150, 215	22–43 V 160–300 mW *	100	0.016, 0.023	N.A.	HeLa	None	Size and amount of transfected elements depend on the voltage, duration, frequency and number of US pulsation. No impact on viability	N.A.
[43]	150, 215	22 V 160 mW *	0.0036	0.5–1.5	N.A.	HeLa	None	Genomic transfection facilitated by US	N.A.

**Table 3 biomedicines-10-01166-t003:** Summary table of the impacts on mammal cells of SAW. (↗: increase in, ↘: decrease in, N.A.: not available, AFM: atomic force microscopy, IDT: interdigital transistor, PDMS: polydimethylsiloxane, *: extrapolation based on the hypothesis that the electrode impedance is at 50 Ω).

Reference	Frequency (MHz)	Intensity or Electrical Power	Duty Cycle (%)	Time	Shear Flow	Device	Cells	Temperature Control	Biological Effects	Hypothesis
[48]	10	65–250 mW	N.A.	N.A.	N.A.	Slanted IDT, LiNbO_3_ chip	Human red blood cells (RBC) RBC infected by the malarial parasite *Plasmodium falciparum*	None	Enrichment, separation of the cells depending on their pathological state	Cell density impacts their displacement with the shear flow
[7]	77–164	80–1000 mW cm^−2^ up to 13.6 mW	100 or 0.00077	5 min–27 h	N.A.	LiNbO_3_ chip covered with a SiO_2_ layer (= substrate), PDMS well	Madin–Darby canine kidney (MDCK-II) Human osteosarcoma sarcoma osteogenic (SaOs-2) Human embryonic kidney (T-REx-293)	Estimated rise of 2.4 °C	Wound healing ↗ cell migration ↗ cell proliferation	Direct mechanical stimulation > flow field, or electrical field
[49]	101–204	380 mW	100	seconds	N.A.	4 IDT, LiNbO_3_ chip	Human lymphocytes RBC infected by the malarial parasite *Plasmodium falciparum*	Thermally controlled chamber	Patterning of spatially isolated individual cells in an acoustic field defined in 2D	N.A.
[50]	48.8	467 mW	2.5	48 h	Shear stress 120–280 mN m^−2^ Shear velocity 600 ± 250 μm s^−1^	LiNbO_3_ chip, titanium substrate, PDMS well	Human monocytes (U-937)	Rise ≤ 0.5 °C	↗ cell proliferation (+36%)	Shear stress linked to SAW has a more positive impact than stirring
[51]	14	Up to18 V, 59.3 mW cm^−2^ and 0.23 µW for a single cell (400 µm^2^) order of magnitude up to 100 mW *	100	4–8 h	Velocity up to 56 µm s^−1^, shear stress 3.8 mPa	LiNbO_3_ chip, glycerol as a coupling liquid with the PDMS cell culture chamber	Mouse embryonic fibroblasts (NIH-3T3)	Feedback loop to maintain the temperature of the medium flow	Cell migration first enhanced, then suppressed as the intensity rose No reduction in cell viability Thicker actin bundles	Cell orientation alignment along the propagating wave, high traction forces activated the Rho signaling pathway
[52]	160	631 mW	100	60 min	Shear rate distribution 1750–6900 s^−1^	Gold IDT, LiNbO_3_ chip, a cylindrical PDMS chamber on top filled with culture medium, cells attached to a titanium implant on top	SaOs-2	Temperature maintained at 37 °C, no precision	Correlation between shear flow and cell detachment from an implant	Cell density plays a key role
[53]	19.35	325–575 mW	100	10 s	Velocity 0–9 mm s^−1^	LiNbO_3_ chip, titanium layer, aluminum substrate,	none	/	↗ penetration rate into a porous scaffold	N.A.
[54]	161–171	31.6 mW	N.A.	>330 µs per pulse	N.A.	Gold and titan LiNbO_3_ chip, covered with glass, PDMS microchannel device	Mouse melanoma cells (B16F10)	None.	Sorting rate of 3000 cells s^−1^ depending on their fluorescence (Calcein-AM)	N.A.
[55]	196.7	1 mW 10–20 kPa	100	3–10 min	N.A.	Quartz (SiO_2_) chip, cells suspended in glycerin, SU-8 microprobe	Chondrosarcoma (JJ012) Breast cancer cells (MDA-MB-231, SKBR3, MCF7)	None	US velocity measurement for single cell analysis 10^6^ sensitivity in elasticity compared to AFM	Cell elastic moduli is a possible biomarker for aggressiveness or metastatic potential
[56]	132	55–500 mW	100	100 s	Velocity 0.42–1.80 m s^−1^ Shear stress 0.01–0.045 Pa	Concentric gold IDT, LiNbO_3_ chip	Untreated, and non-infected human RBC Glutaraldehyde- treated RBC RBC infected by the malarial parasite	None	Cell detachment behavior was different according to the RBC state of infection.	Specific mechanotransduction might be a biomarker
[57]	159	2–4 mW	100	48 h	N.A.	LiNbO_3_ chip, SiO_2_ substrate, PDMS well	SaOs-2	Rise ≤ 0.32 °C	↗ wound healing as a function of US intensity no significant necrosis no preferred direction for migration/proliferation	Unclear if the effect is due to mechanical or electrical stimulation, or a combination of both
[58]	N.A.	316–501 mW	100	0–60 min	Shear flow 2 Pa	LiNbO_3_ chip, titanium substrate	SaOs-2	Thermally controlled chamber	No significant impact on cell adhesion, when T ≤ 37 °C	Decrease in cell adhesion is due to increase in temperature or decrease in pH
[8]	38.74	125.6 mW	80	2 h	N.A.	Two circular IDT (and two straight IDT for SSAW), LiNbO_3_ chip, covered with Al, and PDMS channels	Human glioma cell lines (U87) Rat RBC	None	Cell sorting depending on their virulence	Sorting of particles is dependent on their size

## Data Availability

Not applicable.

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
