# Peer review of "Biological Effects and Applications of Bulk and Surface Acoustic Waves on In Vitro Cultured Mammal Cells: New Insights"

_biomedicines, 2022, doi:10.3390/biomedicines10051166_

Round 1

Reviewer 1 Report

The MS ´Biological effects and applications of bulk and surface acoustic 2
waves on in vitro cultured mammal cells: new insights´ is very interesting and well written. 

The organization of the MS is very good and its with the flow of reading. The authors explained the low and high US wave effects and explained the pros and cons of both of them in the final figure. 

However, I was wondering if the author can add any study that reaches the clinical level that could be more interesting for the reader. 

The addition of a graphical abstract figure including and explaining the elements in this field could enhance the review.  

Author Response

Response to Reviewer 1 Comments

Point 1: The MS ´Biological effects and applications of bulk and surface acoustic 2 waves on in vitro cultured mammal cells: new insights´ is very interesting and well written. The organization of the MS is very good and its with the flow of reading. The authors explained the low and high US wave effects and explained the pros and cons of both of them in the final figure.

However, I was wondering if the author can add any study that reaches the clinical level that could be more interesting for the reader.

Response 1: Thank you for this suggestion. We have added a paragraph at the end of the review “7. Perspectives in biomedicines” that summarizes recent clinical trials and feasability studies of acoustic waves for biomedical applications.

Point 2: The addition of a graphical abstract figure including and explaining the elements in this field could enhance the review. 

Response 2: A graphical abstract has been drawn, and we hope it can indeed guide the future readers.

Reviewer 2 Report

In their review entitled «Biological effects and applications of bulk and surface acoustic waves on in vitro cultured mammal cells: new insights», Figarol et al. have processed a large amount of data. The results have been classified and presented in tables and figures.

Unfortunately, the manuscript is prepared with a number of misleading errors in Figure citations and in formatting Equations (dots are used instead of multiplication sign, while the indices are not formatted in superscript). These issues must be corrected, since they hinder the perception and comprehension of the manuscript.

The authors are encouraged to use the terminology more accurately. So, in Figure 2 caption, "stimulation" is mentioned, but it it more correct to mention an impact, which may (or may not) lead to stimulation.

In general, I have two major concerns:

  1. When the authors consider the impact of US on cells, they do not differentiate the cells from their environment. Obviously, the action on the cells in culture medium is performed. So, is this action on the cells direct or indirect?

For instance, in the first paragraph of Section 3.1, formation of oxygen singlets is mentioned. Where are they formed?

It is very important to take into account the environment of the cells. The authors discuss the applications of the US in biotechnology, medical diagnostics, and therapy. They, however, do not indicate the vectors of contact between the studies performed with cell cultures, and the possibility of application in practice — for instance, for killing cancer cells in the body.

  1. Which parameters of the US can be useful in clinical practice with respect to treatment of patients? In the body, malignant cells are obviously surrounded by other (normal) cells. Reflection and absorption of the US take place, thus causing thermal effect of the US. How does skin and adipose tissue layer influence the US parameters?

In my opinion, the above-mentioned information should be given in the manuscript in order to to enhance the overall merit of this review. Alternatively, the data, summarized in the review, can be presented to be useful for only the biotechnology, while their value for the biomedicine should not be emphasized.

Based on the above, my overall recommendation is major revision.

Author Response

Response to Reviewer 2 Comments

Point 1: In their review entitled «Biological effects and applications of bulk and surface acoustic waves on in vitro cultured mammal cells: new insights», Figarol et al. have processed a large amount of data. The results have been classified and presented in tables and figures.

Unfortunately, the manuscript is prepared with a number of misleading errors in Figure citations and in formatting Equations (dots are used instead of multiplication sign, while the indices are not formatted in superscript). These issues must be corrected, since they hinder the perception and comprehension of the manuscript.

Response 1: We thank reviewer 2 for his thorough reading. Indeed, some automatic cross-references were impaired by the last formatting. Those references were corrected. In accordance to the recommandations for international units, we have replaced the dots between units by spaces (example: W.cm-2 became W cm-2).

Point 2: The authors are encouraged to use the terminology more accurately. So, in Figure 2 caption, "stimulation" is mentioned, but it it more correct to mention an impact, which may (or may not) lead to stimulation.

Response 2: We have removed the term “stimulation” from the table titles, as it could lead to confusion. It was intended as an electrical stimulation, but could be understood as a cellular stimulation.

Point 3: In general, I have two major concerns: 1) When the authors consider the impact of US on cells, they do not differentiate the cells from their environment. Obviously, the action on the cells in culture medium is performed. So, is this action on the cells direct or indirect?
For instance, in the first paragraph of Section 3.1, formation of oxygen singlets is mentioned. Where are they formed? It is very important to take into account the environment of the cells.

Response 3: This is a valuable comment. In paragraph 3.1., we have precised that the cavitation phenomenon is happening in the external medium, with a new reference (Miller et al.) that elaborated on this. It was also precised line 245 that the microbubbles resulting from these cavitations are equally formed in the culture medium. In paragraph 5.1., it was added that the changes in pH and temperature were also measured in the culture medium and not intracellular values. In pargraph 6, that sumed up the need for standardization between the studies, those precisions were mentioned once again for clarification.

Point 4: The authors discuss the applications of the US in biotechnology, medical diagnostics, and therapy. They, however, do not indicate the vectors of contact between the studies performed with cell cultures, and the possibility of application in practice — for instance, for killing cancer cells in the body. 2) Which parameters of the US can be useful in clinical practice with respect to treatment of patients? In the body, malignant cells are obviously surrounded by other (normal) cells. Reflection and absorption of the US take place, thus causing thermal effect of the US. How does skin and adipose tissue layer influence the US parameters?

Response 4: We have added a paragraph at the end of the review to answer to this comment and the reviewer 1 second comment. The new part “7. Perspectives in biomedicines” summarizes recent clinical trials and feasability studies of acoustic waves for biomedical applications. When possible, we have stated the parameters used in these studies. However, the data for ongoing clinical trials are not public, and the feasability studies are most often still in the in vitro stage. It is thus quite challenging to retrieve this information. We emphasized that each cell type reacts differently to acoustic waves, expecially cancerous or malignant versus healthy cells. Some preliminary studies tried to use this to focalize the impacts of ultrasounds, some other studies have chosen to adjust the acoustic waves parameters to reach the same goal. The cell environment will surely modify the action of acoustic waves, but the work on this field is still exploratory. Further mechanobiological studies and transcriptomic analysis would be of interest to underlined the differences in responses to US exposure according to the cell type and state.

Round 2

Reviewer 2 Report

The authors have done a good job, and have properly addressed my concerns, and the manuscript can now be accepted for publication.

The only point is the Figure 2 caption, which has not been corrected. The authors are encouraged to correct this point, and to fix superscript formatting issues (L. 147 and 184).

Sincerely,

The reviewer